# Photonic Crystal Fiber Plasmonic Sensor Based on Dual Optofluidic Channel

**DOI:** 10.3390/s19235150

**Published:** 2019-11-25

**Authors:** Nan Chen, Min Chang, Xinglian Lu, Jun Zhou, Xuedian Zhang

**Affiliations:** 1School of Optoelectronic Information and Computer Engineering, University of Shanghai for Science and Technology Shanghai Key Laboratory of Modern Optical System, 516 Jungong Rd, Shanghai 200093, China; 181560053@st.usst.edu.cn (N.C.); changmin@usst.edu.cn (M.C.); 151360021@st.usst.edu.cn (X.L.); 151360026@st.usst.edu.cn (J.Z.); 2Shanghai Key Laboratory of Molecular Imaging, Shanghai University of Medicine and Health Sciences, Shanghai 201318, China

**Keywords:** photonic crystal fiber, surface plasmon resonance, fiber sensor, fabrication

## Abstract

A hexagonal photonic crystal fiber (PCF) sensor with a dual optofluidic channel based on surface plasmon resonance (SPR) effect is proposed. The sensor characteristic is numerically explored by software integrated with the finite element method (FEM). The numerical results show that, when the analyte refractive index (RI) varies from 1.32 to 1.38, high linearity between resonance wavelength and analyte RI is obtained and the value of adjusted R^2^ is up to 0.9993. Simultaneously, the proposed sensor has maximum wavelength sensitivity (WS) of 5500 nm/RIU and maximum amplitude sensitivity (AS) of 150 RIU^−1^, with an RI resolution of 1.82 × 10^−5^ RIU. Besides, owing to a simple structure and good tolerance of the proposed sensor, it can be easily fabricated by means of existing technology. The proposed sensor suggests promising applications in oil detection, temperature measurement, water quality monitoring, bio-sensing, and food safety.

## 1. Introduction

Surface plasmon polariton [1] (SPP) has attracted increasingly more attention recently. Generally, when light incidents to a metal–dielectric interface, the interaction between light and matter is produced at this interface with significant energy loss; as long as the phase matching condition (PMC) is satisfied, the plasmonic resonance coupling state can be maintained. The above phenomenon is known as “surface plasmon resonance” (SPR) [2,3]. In the last decades, SPR technology has been applied widely in various fields, such as food safety, liquid and gas detection, bio-sensing, drug detection, and so on [4,5,6,7,8]. Specially, by integrating the SPR effect, photonic crystal fiber (PCF) [9,10] can play a more important role in sensors, owing to its small size and flexible structure. By varying PCFs’ dimension and arrangement of the pores, it is possible to manipulate the sensitivity and resonance peak of the required PCF-SPR sensors.

Nowadays, increasingly more novel designs of PCF-SPR sensor are being designed and numerically analyzed. Some typical designs are as follows: Dash et al. proposed a biosensor based on PCF-SPR that showed refractive index (RI) sensitivity of 2000 nm/RIU with a resolution of 5 × 10^−5^ RIU [11]. In 2014, Otupiri et al. proposed a novel birefringent PCF biosensor constructed on the SPR effect; for spectral interrogation (SI), the sensor resolution values yielded 5 × 10^−5^ RIU for ${\mathrm{HE}}_{11}^{x}$ modes and 6 × 10^−5^ RIU for ${\mathrm{HE}}_{11}^{y}$ modes; for amplitude interrogation (AI), the sensor amplitude sensitivities (ASs) of 3 × 10^−5^ RIU for ${\mathrm{HE}}_{11}^{x}$ modes and 4 × 10^−5^ RIU for ${\mathrm{HE}}_{11}^{y}$ modes were obtained [12]. In 2015, selecting gold (Au) and silver (Ag) as the SPR activity metal, Fan et al. proposed two kinds of novel plasmonic RI sensors; their numerical results showed average sensitivities of 7040 nm/RIU (Au) and 7017 nm/RIU (Ag) with high linearity [13]. In 2016, Rifat et al. designed a novel PCF biosensor based on SPR phenomena, in which the sensing Au layer was wrapped outside the PCF; by using the wavelength interrogation (WI) mode, the designed sensor showed a maximum sensitivity of 14,000 nm/RIU and a resolution of 1 × 10^−4^ RIU [14]. In 2017, Wu et al. designed a side-polished D-shaped PCF-SPR sensor by a combination of experiment and simulation; the sensitivity was up to 21,700 nm/RIU in the RI range of 1.33–1.34 [15]. Liu et al. proposed a PCF-SPR sensor with two open-ring channels; their sensor showed an average spectral sensitivity of 5500 nm/RIU and a maximum resolution of 7.69 × 10^−6^ RIU [16]. 

According to the above, most of the PCF-SPR sensors with excellent performance are mainly employed in RI detection, ranging from 1.32 to 1.38 for bio-sensing, and the PCF structure and processing techniques for designing these sensors vary. Therefore, we try to introduce the dual optofluidic channel in PCF and implement the proposed sensor with detection ranging from 1.32 to 1.38, which can be applied in practical fields. In this paper, we propose and numerically analyze a PCF-SPR sensor with a simple structure. By investigating the transmission loss for different sizes of the innermost four pores, Au-filled central pore diameter of the PCF, and RI coefficients of the analytes, we find the structural parameters have significant effects on the resonance coupling between the polarized core mode and SPP mode. In particular, the RI of filling material will be helpful for improving the sensitivity of PCF-SPR sensors. The proposed sensor can be fabricated easily by employing the standard stack and draw PCF fabrication method.

## 2. Numerical Modeling 

Figure 1 displays the schematic cross section of this PCF-SPR sensor. We propose a three-layer hexagonal lattice with an Au-filled central pore and the innermost enlarged four pores. Optofluidic channels are on both sides of the Au wire and analytes are introduced in the pore channels. An energy coupling effect can be generated by the SPP mode generated between Au in the central pore and the fundamental core mode of the filled analytes. In the proposed PCF, all pores are of a circular shape. The diameter of the PCF is 15 μm and the pore-to-pore distance (pitch) is *Ʌ* = 2 μm. The diameter of pores of the two outer layers and analyte-filled pores is represented by *d_1_*, the innermost enlarged pore diameter is represented by *d_2_*, and diameter of the central Au-filled pore marked in orange is represented by *d_Au_*. The RI detection range for the proposed sensor is from 1.32 to 1.38 and the RI of analyte marked in blue is represented by *n_a_*. Fused silica is selected as the substrate material for PCF; its RI Sellmeier equation [17,18] is described as follows:(1)$$n^{2}\left(\lambda \right)=1+B_{1}\lambda^{2}{\left(\lambda^{2}-C_{1}\right)}^{-1}+B_{2}\lambda^{2}{\left(\lambda^{2}-C_{2}\right)}^{-1}+B_{3}\lambda^{2}{\left(\lambda^{2}-C_{3}\right)}^{-1},$$
where *B_1_* = 0.696163, *B_2_* = 0.4079426, and *B_3_* = 0.8974794; *C_1_* = 0.0046791 ${\mu m}^{2}$, *C_2_* = 0.0135121 ${\mu m}^{2}$, and *C_3_* = 97.93400 ${\mu m}^{2}$. The dielectric constant of Au filled in the central pore can be defined by the Drude–Lorentz (LD) model [19,20,21,22,23,24]:(2)$$\varepsilon \left(\omega \right)=\varepsilon_{1}+i\varepsilon_{2}=\varepsilon_{\infty }-\frac{\omega_{p}^{2}}{\omega \left(\omega +i\omega_{c}\right)},$$
where $\varepsilon_{\infty }$ is the dielectric constant of Au at a high frequency and its value is 9.75. The plasma frequency $\omega_{p}$ and the scattering frequency of electron $\omega_{c}$ are $1.36\times 10^{16}$ Hz and $1.45\times 10^{14}$ Hz, respectively.

The finite element method (FEM) [25,26] is used to calculate the results with COMSOL5.2 software platform. When incident light propagates along the axial direction, different modes will distribute on the fiber end face, and modal analysis can be performed. Around the PCF structure, selecting the perfectly matched layer [27] (PML) as the boundary condition (BC) for absorbing scattered energy to improve accuracy of calculation and the transmission loss spectral can be characterized by confinement loss [28,29,30], which is expressed as follows: (3)$$\alpha \left(\lambda ,\;n_{a}\right)=8.686\times \frac{2\pi }{\lambda }\times Im\left(n_{eff}\right)\times 10^{3}\left(dB/mm\right),$$
where *λ* and *Im(n_eff_)* represent the operating wavelength in vacuum in μm and imaginary part of *n_eff_* for the fundamental mode (FM), respectively. When it comes to sensitivity, wavelength sensitivity (WS) and amplitude sensitivity (AS) are generally referred to. By means of the transmission loss, the proposed sensor performance parameters can be calculated. The WS and AS can be given by the following:(4)$$S\left(\lambda \right)=\frac{∆\lambda_{peak}}{∆n_{a}}\left(nm/RIU\right),$$
and
(5)$$S_{A}\left(\lambda \right)=\frac{1}{\alpha \left(\lambda ,n_{a}\right)}\frac{\partial \alpha \left(\lambda ,n_{a}\right)}{\partial n_{a}}\left(RIU^{-1}\right),$$
respectively, where $∆\lambda_{peak}$ denotes the resonance wavelength changes and $∆n_{a}$ denotes variations in RI. Furthermore, the PML thickness is 2.5 μm; discretizing the whole cross section with fine triangular mesh elements of 130,897. The source is an electromagnetic hybrid wave that is perpendicular to the x–y plane, and the full-wave analysis method is utilized to investigate the mode distributions.

## 3. Numerical Results and Discussions 

### 3.1. Dispersion Relation

The proposed PCF-SPR sensor operates on the basis of the interaction between evanescent field and central Au wire. The filling Au in the central pore between two analyte-filled cores can significantly enhance the birefringence and polarization-dependent loss for core FMs, resulting in visible differences of resonance peaks. As shown in Figure 2, the excited plasmonic modes are named as follows: (a) 0-SPP mode; (b) 1-SPP1 mode; (c) 1-SPP2 mode; (d) 2-SPP1 mode; (e) 2-SPP2 mode; (f) 3-SPP1 mode; and (g) 3-SPP2 mode [31]. The 1-SPP, 2-SPP, and 3-SPP modes usually have two degenerate states. In the universal band, the energy generated by resonance between FM and the 2-SPP mode is greater than that by resonance between FM and other SPP modes [32], so the 2-SPP mode is usually utilized to generate sensing signals. For the dual-core PCF-SPR sensor, the even mode and odd mode for x- and y-polarization (Figure 2h–k) are excited simultaneously when incident light propagates along the z direction, and most of the energy is gathered in the dual-core. According to the calculated mode distributions, there are usually four states; they are the x-polarized even mode (x-EM) (Figure 2h); x-polarized odd mode (x-OM) (Figure 2i); y-polarized even mode (y-EM) (Figure 2j); and y-polarized odd mode (y-OM) (Figure 2k), respectively. Generally, the parity can be distinguished based on the direction of the arrow, while the even modes possess similar properties and both arrows point in the same direction, the odd modes have phase difference π between two cores with two arrows in opposite directions [33,34]. Both even and odd polarized modes perform resonance coupling with 2-SPP modes being investigated. There are two cases of complete coupling resonance situations, as shown in Figure 3. The PMC of the sensor is satisfied and *Re(_neff_)* and 2-SPP mode RI are equal at a resonant wavelength. In the process, the excited free electron oscillations owing to irradiation cause surface plasmon wave (SPW) in the metal–dielectric interface, so the evanescent field can be enhanced and the *Im(n_eff_)* curve shows a sharp peak. There is only one distinct characteristic peak in the detected bands and the energy is close to 0 at the wavelengths of both sides away from the SPR peak, so it is very suitable for sensing applications. Additionally, it can be clearly seen that the sharpness of y-OM is higher than that of x-OM; in other words, the confinement loss of the former is stronger than that of the latter in this sensor. This can be explained in that the transmission loss is proportional to *Im(n_eff_)*.

### 3.2. PCF-SPR Sensor Performance

On the basis of the sensor’s mode profile and satisfaction of PMCs, as shown in Figure 2 and Figure 3, the calculated loss spectra changes owing to different structural parameters are shown in Figure 4. In Figure 4a, *n_a_* is temporarily set as 1.35, with structural parameters of *Ʌ* = 2 μm, *d_1_* = 0.8 μm, and *d_Au_* = 0.8 μm, whereas the innermost enlarged pore diameter *d_2_* is taken as 1.4 μm, 1.6 μm, and 1.8 μm, respectively. With the increment of *d_2_*, the resonance peaks for x- and y-polarization produce a slight blue shift and confinement loss also decreases in this process. In addition, it can be clearly seen that the confinement loss reduction scale of the resonance peak for x-polarization is larger than that of y-polarization. Therefore, although both of the resonance peaks for x-polarization and y-polarization have decreased, the difference in energy between the two resonance peaks becomes more and more obvious. As a result, *d_2_* = 1.8 μm is adopted in order to distinguish the interference between the two confinement losses. In Figure 4b, making structural parameters *Ʌ* = 2 μm, *d_1_* = 0.8 μm, and *d_2_* = 1.8 μm, Au-filled central pore diameter *d_Au_* is taken as 0.6 μm, 0.7 μm, and 0.8 μm, respectively. It can be intuitively found that the resonance wavelength moving is particularly sensitive to changes in *d_Au_*. As *d_Au_* increases, the resonance peaks for x- and y-polarization produce obvious red shifts and the confinement loss for y-polarization increases drastically; thus, the energy of the two polarized directions can also be clearly distinguished. On the basis of the above analysis, it is suitable to choose the structural parameters of *Ʌ* = 2 μm, *d_1_* = 0.8 μm, *d_2_* = 1.8 μm, and *d_Au_* = 0.8 μm to design the sensor. 

In the following, y-OM is employed to evaluate the performance of this sensor. The real part of the fundamental dual core mode RI is influenced strongly by the analyte RI, which determines the phase matching wavelength (PMW) between the core mode and the SPP mode. By varying the analyte RI from 1.32 to 1.38, a series of loss curves can be obtained, as shown in Figure 5a. The blue shift of resonance wavelengths of the y-polarized mode can be observed as *n_a_* increases. The reason is that the RI of the core mode increases and that of the SPP mode changes only slightly, which leads to a blue shift of the PMW. At the same time, the resonance peak shows an increasing trend. Figure 5b shows the loss spectra of FM for different analyte RI with a single filled pore. Schematic diagram and mode field distributions are shown in the inset. Similarly, y-component FM is considered. The results in Figure 5b cannot be compared with those in Figure 5a, because the loss spectra changes irregularly, the resonance peak increases firstly and then remains unchanged with the increase of *n_a_*, and the operating wavelength range is more narrow. According to Figure 5, we investigated the relationship between analyte RI and resonance wavelength. Adopting Excel, linearly fitting discrete points were performed; the linear fitting curves are shown in Figure 6 and the obtained linear fitting formulas are shown as follows:(6)$$\lambda_{1}=-\;0.975n+0.3928\;\left(\mu m\right),\left(1.32\leq n\leq 1.38\right),$$
(7)$$\lambda_{2}=-\;0.825n+0.1856\;\left(\mu m\right),\left(1.32\leq n\leq 1.38\right),$$
where *n* represents analyte RI in RIU. The quality of linearity can be judged by evaluating the adjusted *R^2^* (coefficient of determination) [35]. By calculation, the values of adjusted *R^2^* for *λ_1_* and *λ_2_* are 0.9993 and 0.9412, respectively, which shows that the results of the former possess higher linearity compared with those of the latter.

In this sensor, a slight change in RI can cause significant spectra shifts. As shown in Figure 5a, the resonance wavelength changes regularly with the increase of analyte RI at intervals of 0.1, which displays the wavelength dependence of the loss spectra owing to the change of analyte RI. The WS (also known as spectral sensitivity) is one of the important sensing performance parameters [36], as shown in Equation (4). Usually, $∆n_{a}$ is assumed to be 0.01. In this sensor, the maximum $∆\lambda_{peak}$ is about 55 nm and the maximum spectral sensitivity is 5500 nm/RIU; combining all the cases, the average $∆\lambda_{peak}$ is about 10.2 nm and the average spectral sensitivity is 1020 nm/RIU. Usually, a wavelength resolution of 0.1 nm is assumed, and the RI resolution can reach 1.82×10^−5^ RIU.

Besides, the sensing AS is another important parameter [37], which can be calculated by Equation (5). The AS curves can be obtained by varying the analyte RI. Figure 7 shows that the AS increases gradually with the increase of analyte RI. Obviously, there is still better performance in the case of double pores filling. The maximum AS of this sensor is approximately 150 RIU^−1^, when the analyte RI is 1.37. Therefore, this sensor has both high linearity and high sensitivity. Table 1 shows a comparison of our designed sensor with that of the existing typical designs. The magnitude of the maximum spectral sensitivity, the maximum amplitude sensitivity, the resolution, and the linearity for this proposed sensor can reach the performance level of the peers, which proves that our designed sensors possess good performance.

## 4. Fabrication Consideration

Nowadays, the stack-and-draw method [43,44], extruded method, and gas producing method are common fabrication methods for PCF. Among them, the stack-and-draw method is relatively mature and varieties of the PCF structures can be drawn stably. Therefore, the stack-and-draw method is preferred for PCF fabrication. Firstly, the capillaries are usually stacked into a hexagonal structure, referred to in Figure 1, and the central capillaries are removed to form defects; these stacked capillaries are then placed in a thin-walled quartz tube of a suitable dimension, and a capillary is used to fill the peripheral voids to form a high-quality preform. Then, the preform is put into the fiber drawing tower, and the PCF with the desired dimensions can be drawn out. Secondly, Au wire is filled into the central pore of the PCF. All the pores except the central one are blocked with glue at one end of the PCF; Before filling Au wire, the appropriate internal pressure and precise heating temperature to soften glass are necessary for PCF, and then molten Au is pumped into the central pore, yielding the desired Au wire lengths. Eventually, after cleaving off the unfilled section, the desired PCF sensor is achieved [45]. Therefore, the PCF-SPR sensor we designed is simple in structure and can be completely fabricated using existing technology.

Finally, the design tolerance is considered. In the process of fiber fabrication, it is difficult to accurately control the size of the pores and pitch. The current technology can guarantee the error to be controlled at about 1% [46]. For the proposed sensor, ±3% variation of *Ʌ* and *d_1_* in the structure is investigated, as shown in Figure 8. Some slight changes in the loss and resonance wavelengths can be found. Especially, a change occurred in the case of −3% of Ʌ. This could be explained in that the pore spacing decreases and the pores are denser as Ʌ decreases, which shows a slightly more pronounced modulation for wavelength and energy. With the development of modern technology, precise control for Ʌ and d_1_ can be achieved, and these effects of the changes will be expected to be controlled.

## 5. Conclusions

In conclusion, a practically simple, hexagonal PCF-SPR sensor with dual optofluidic channel is proposed and numerically investigated, which is characterized by Au-filled, innermost four enlarged pores and double pore-filled analytes. The FEM is employed to analyze the sensing characteristic by optimizing the structural parameters in PCF. After calculation, the results show that when *Ʌ* = 2 μm, *d_1_* = 0.8 μm, *d_2_* = 1.8 μm, and *d_Au_* = 0.8 μm, the structure is suitable for sensing. According to the designs of Shuai et al. [47] and Qin et al. [48], they used multiple filling cores to detect RI larger than that of the fiber substrate and all achieved very good sensing performances. Although our paper investigates the RI range of 1.32–1.38, our sensors are also suitable for larger RI ranges. We believe that the proposed sensor will play a significant role in practical applications in the future. 

## Figures and Tables

**Figure 1 sensors-19-05150-f001:**
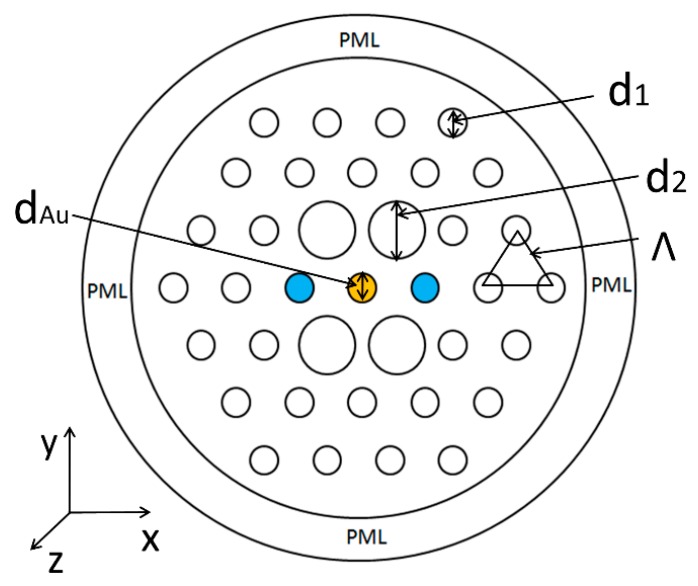
Cross-section diagram of the proposed photonic crystal fiber (PCF)-surface plasmon resonance (SPR) sensor.

**Figure 2 sensors-19-05150-f002:**
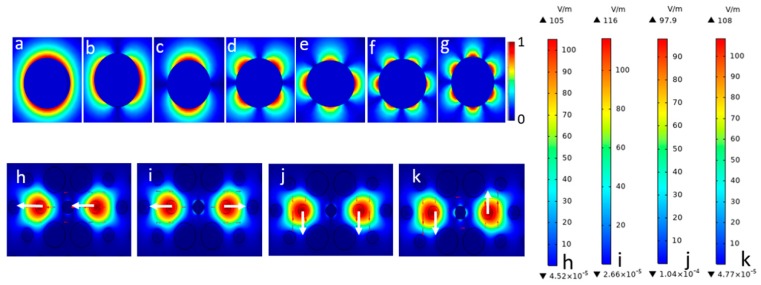
The electric field distributions. (**a**) 0-surface plasmon polariton (SPP) mode; (**b**) 1-SPP1 mode; (**c**) 1-SPP2 mode; (**d**) 2-SPP1 mode; (**e**) 2-SPP2 mode; (**f**) 3-SPP1 mode; and (**g**) 3-SPP2 mode. (**h**–**k**) The distributions of the even and odd supermodes (the arrows indicate the instantaneous electric field direction).

**Figure 3 sensors-19-05150-f003:**
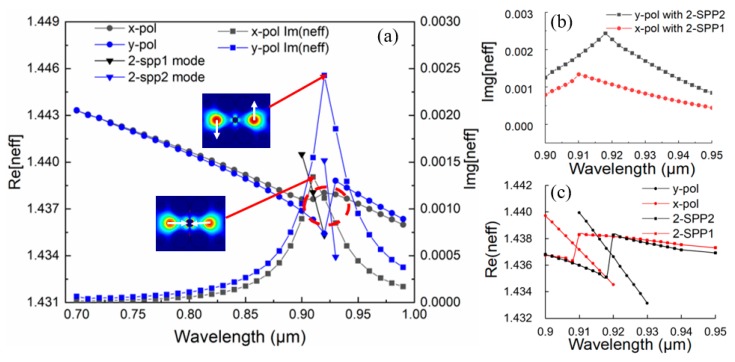
(**a**) The effective refractive index (RI) of the odd polarized core modes and SPP mode, and the imaginary part of RI for the proposed surface plasmon resonance (SPR) sensor with *Ʌ* = 2 μm, *d_1_* = 0.8 μm, *d_2_* = 1. 4 μm, *d_Au_* = 0.8 μm, and *n_a_* = 1.35; (**b**) *Im[n_eff_]* partial enlargement at wavelength ranging from 0.9 to 0.95; (**c**) *Re[n_eff_]* partial enlargement at wavelength ranging from 0.9 to 0.95 corresponding to the red dotted line.

**Figure 4 sensors-19-05150-f004:**
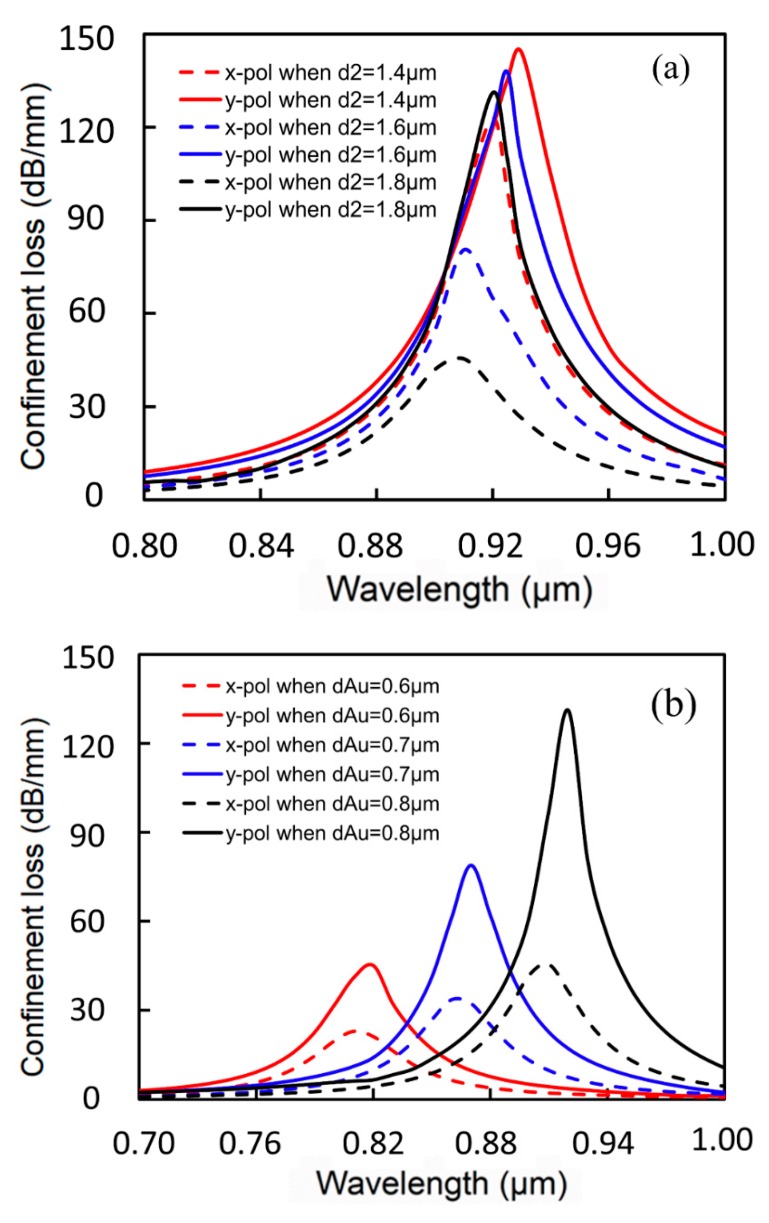
(**a**) Loss spectra of the fundamental mode (FM) for different sizes of the innermost enlarged pore d_2_; (**b**) loss spectra of the plasmon peaks for different diameters of central Au-filled pore d_Au_.

**Figure 5 sensors-19-05150-f005:**
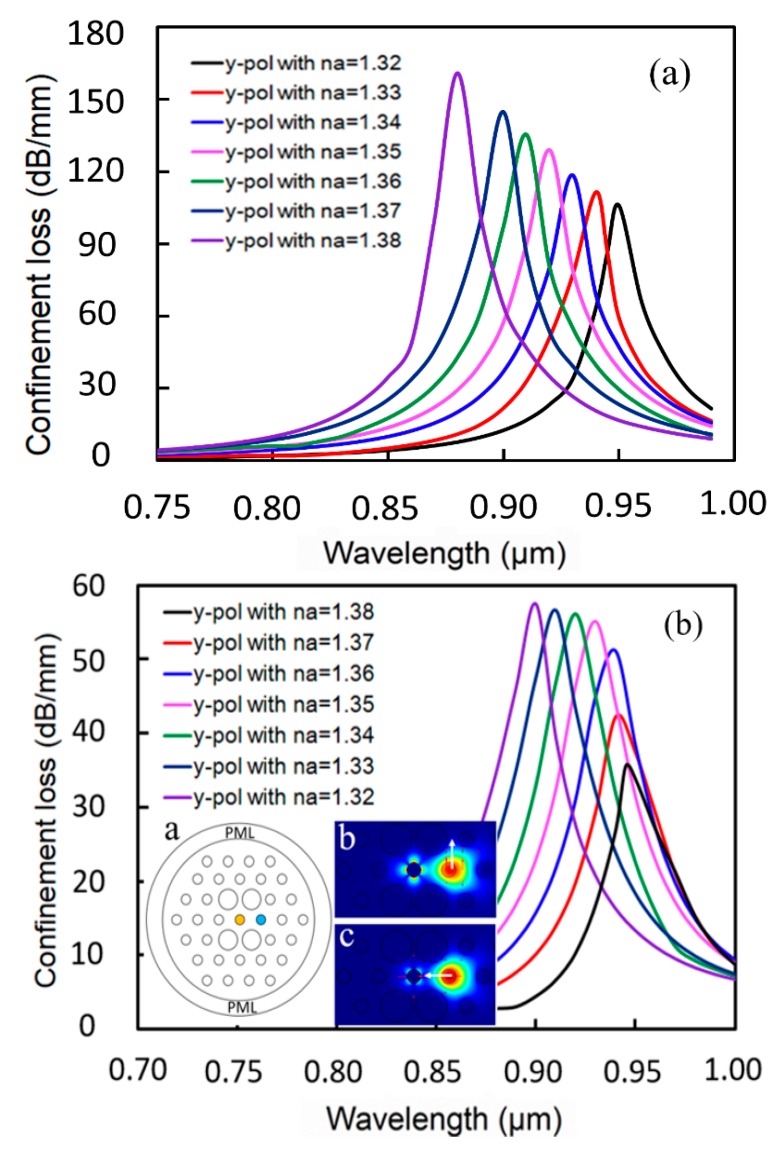
Loss spectra of FM in the proposed sensor structure for different analyte RI between 1.32 and 1.38 with (**a**) dual channel and (**b**) single channel.

**Figure 6 sensors-19-05150-f006:**
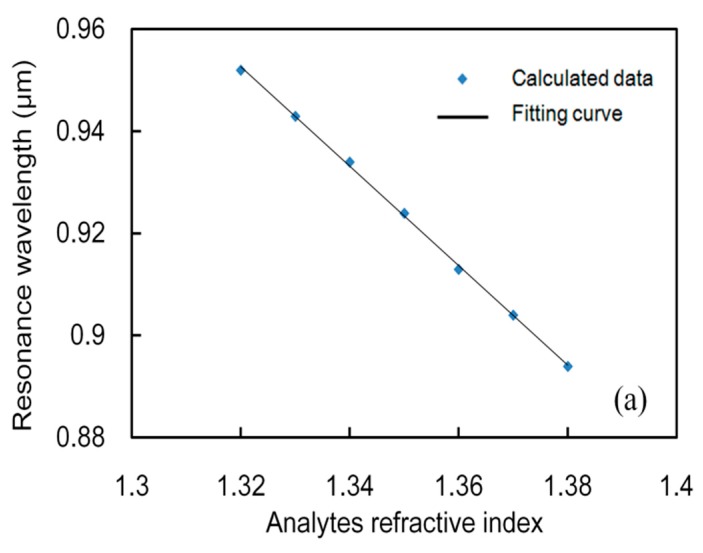
Linear fitting curve of the FM resonance wavelength versus analyte RI from 1.32 to 1.38 with (**a**) dual channel and (**b**) single channel.

**Figure 7 sensors-19-05150-f007:**
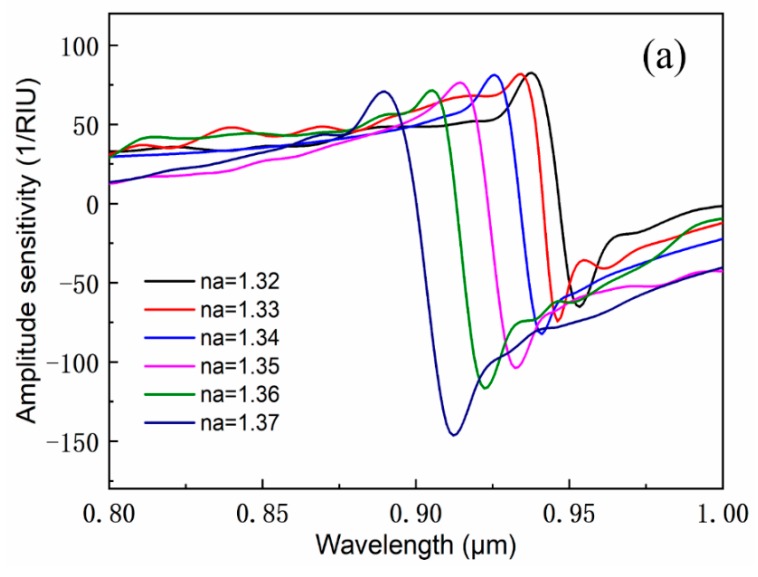
Amplitude sensitivity of the proposed sensor for different RI of the filled analytes with *Ʌ* = 2 μm, *d_1_* = 0.8 μm, *d_2_* = 1.8 μm, and *d_Au_* = 0.8 μm with (**a**) dual channel and (**b**) single channel.

**Figure 8 sensors-19-05150-f008:**
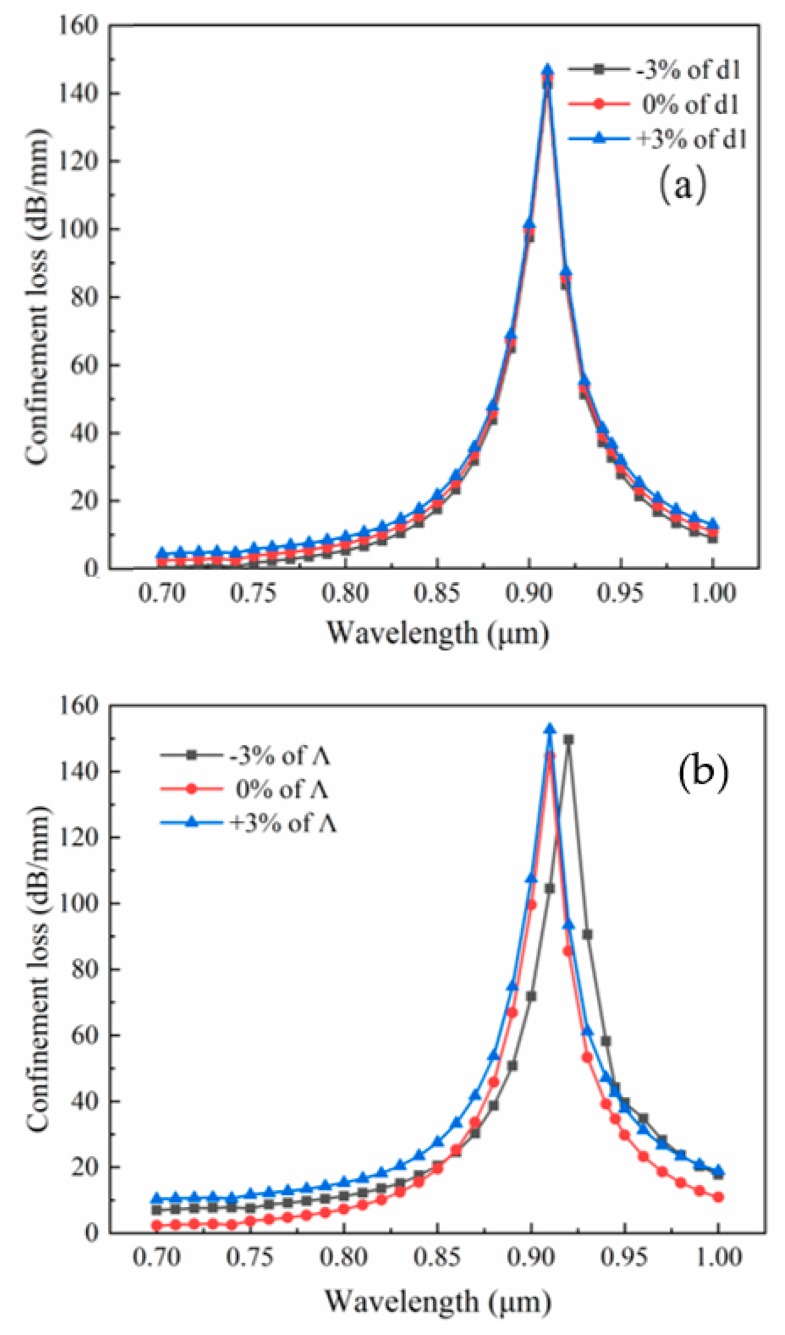
Loss spectra for variation of (**a**) *d_1_* of ± 3%; (**b**) *Λ* of ±3% with *Ʌ* = 2 μm, *d_1_* = 0.8 μm, *d_2_* = 1.8 μm, *d_Au_* = 0.8 μm, and *n_a_* = 1.37.

**Table 1 sensors-19-05150-t001:** Performance comparison between the proposed sensor and prior typical sensors.

Ref	Structure Type	Max. Wav. Sens. (nm/RIU)	Max. Amp. Sens. (RIU^−1^)	Resolution (RIU)	Linearity
[38]	Long period gratings	1500	N/A	6.67 × 10^−5^	N/A
[39]	Planar waveguide coupled resonators	182	N/A	5.56 × 10^−4^	N/A
[40]	Capillary ring resonators	800	N/A	1.25 × 10^−4^	N/A
[41]	Dual-polarized	4600	420.4	N/A	0.9720
[5]	Outer coating	4000	320	3.125 × 10^−5^	N/A
[42]	Side polished	5200	N/A	1.92 × 10^−5^	N/A
This paper	Dual optofluidic channel	5500	150	1.82×10^-5^	0.9993

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
