# Peer review of "Photonic Crystal Fiber Plasmonic Sensor Based on Dual Optofluidic Channel"

_sensors, 2019, doi:10.3390/s19235150_

Round 1
Reviewer 1 Report
On Section 2 Numerical modeling, it was not clear what is the RI of analytes, which the authors should make clear. Moreover, I am curious about the guiding mechanism of light in the filled cores. I assume the light is confined in glass and filled core region by total internal reflection on the air-glass interface. However, in practice, this requires RI of analytes being closed with the silica glass. On one hand, if the RI is too small, it forms an anti-waveguide which is practically impossible to couple light in the filled region, though the modes can be found in FEM. In this case, most of light will be confined in the glass region rather than the filled cores. On the other hand, if the RI is comparably larger than the silica glass, the filled core becomes a waveguide itself, the modes can ‘feel’ more sensitively the change of the analytes. I would suggest the author give more details on how fundamental modes changes with RI changes on a large scale (1.3-1.6). 3 the spp mode is not clear shown in the plot. I would suggest another figure with zoom-in from wavelength around 0.9 to 0.95 with more delicate wavelength scanning. Another suggestion would be smaller marker in this plot, the lines should be highlighted. The author should point out what is the baseline loss of the system at wavelengths away from SPR. And I would suggest the author change the y-axis to dB/mm, since it is not easy to find any equipment with 300 dB dynamic range. It won’t be possible to scan out a loss peak with 1-cm fibre.
Author Response
Reviewer #1
Comment 1: On Section 2 Numerical modeling, it was not clear what is the RI of analytes, which the authors should make clear.
Thank you for your comments. The proposed PCF-SPR sensors are mainly used to detect analytes, of which the refractive index changes from 1.32 to 1.38, we have stated it on L70 in Section 2.
Comment 2: Moreover, I am curious about the guiding mechanism of light in the filled cores. I assume the light is confined in glass and filled core region by total internal reflection on the air-glass interface. However, in practice, this requires RI of analytes being closed with the silica glass. On one hand, if the RI is too small, it forms an anti-waveguide which is practically impossible to couple light in the filled region, though the modes can be found in FEM. In this case, most of light will be confined in the glass region rather than the filled cores. On the other hand, if the RI is comparably larger than the silica glass, the filled core becomes a waveguide itself, the modes can ‘feel’ more sensitively the change of the analytes. I would suggest the author give more details on how fundamental modes changes with RI changes on a large scale (1.3-1.6)
Response: Thank you for your comments. Most of the PCF-SPR sensors are mainly employed in RI ranging from 1.32 to 1.38. Actually, we also want to show that the proposed sensor can utilized to detect the changes of low RI (1.32-1.38) for analytes. In our design, light will be confined in the glass region rather than the filled cores as the RI decreases, but the filled analytes still has function of modulation in the sensing process; Our paper emphasizes the refractive index detection range of 1.32-1.38, we can also detect RI changes on a large scale of 1.3-1.6, but because we need to do a lot of calculations and can't finish it in 10 days, so in the revised draft, we provide some literatures to verify that when the RI is comparably larger than the silica glass, the filled core becomes a waveguide itself, the sensor is also sensitive to the change of the analytes. So our sensor is also suitable for detecting larger RI. We have added relevant statements and marked red in Conclusion.
Comment 3: Fig. 3 the spp mode is not clear shown in the plot. I would suggest another figure with zoom-in from wavelength around 0.9 to 0.95 with more delicate wavelength scanning. Another suggestion would be smaller marker in this plot, the lines should be highlighted. The author should point out what is the baseline loss of the system at wavelengths away from SPR.
Response: Thank you for your suggestions. We have updated Fig. 3, and intensively take points in the wavelength range of 0.9-0.95, adding Fig3 (b) and Fig. 3 (c); The baseline loss of the system in our paper is considered as 0.
Comment 4: I would suggest the author change the y-axis to dB/mm, since it is not easy to find any equipment with 300 dB dynamic range. It won’t be possible to scan out a loss peak with 1-cm fibre.
Response: Thank you for your suggestions. We are sorry that we did not consider the practical problems of difficulty with 300 dB dynamic range. We have changed the y-axis to dB/mm in Fig.4, Fig.5 and Fig.8.
Reviewer 2 Report
In their manuscript, N. Chen et al. propose a refractive index (RI) sensor design based on a hexagonal photonic crystal fiber with single central Au-filled pore and dual optofluidic channels. The sensor is capable for measuring small analyte RI variations flowing through the channels by detecting the changes in the propagation constant of the guiding fiber optical mode. The authors present the results of their numerical simulations and claim the proposed sensor has very high spectral sensitivity, the RI resolution and amplitude sensitivity.
I found the paper presented for the review is scientifically sound, and the results in principle may deserve the publication in MDPI Sensors journal. However, in the present form I cannot recommend this paper for the publication because of the English of the paper which should be substantially improved and polished with the help of a native speaker; sometimes it is hard to follow the authors’ logic. Moreover, there are several points that have to be properly addressed by the authors.
Several points that may help to improve the paper readability:
Abstract is a simply replica of the Conclusion section; this is in some way bad manners; The research motivation presented in the Introduction is unclear and should be strengthen; What is “a three-ring hexagonal lattice” (page 2)? May be the authors meant a circularly arranged hexagonal lattice? Here, the plasma frequency and electron frequency are given without units (Hz); More details of the computer simulations will be welcome (i.e., COMSOL version, model dimension and type (full-wave or scattered field), light wave polarization; Page 3, the designations for the confinement loss alpha[loss] are different in eqs. (3) and (5); I did not understand what are the “2-SPP” modes, please explain. How the eigenmodes of different polarizations (even, odd) were distinguished in the simulations? In fig. 3 the first and next to last data in the legend are marked identically. What is the meaning of the symbols (points) in this graph? If they indicate the specific values where the calculations were done so they should become denser in the region of function maximum; In eqs. (6) and (7) the units (um) are missing in the right side; Regarding fig. 8 I do not agree with authors that the tolerance of proposed sensor to the pores pitch variations (lambda) is as good as reported. Indeed, when examining figure 8b one can see that the change in lambda by -3% can produce the changes in the confinement losses of about 40% at 930 nm illumination wavelength! This should be discussed in more details.
Author Response
Reviewer #2
Comment 1: In their manuscript, N. Chen et al. propose a refractive index (RI) sensor design based on a hexagonal photonic crystal fiber with single central Au-filled pore and dual optofluidic channels. The sensor is capable for measuring small analyte RI variations flowing through the channels by detecting the changes in the propagation constant of the guiding fiber optical mode. The authors present the results of their numerical simulations and claim the proposed sensor has very high spectral sensitivity, the RI resolution and amplitude sensitivity. I found the paper presented for the review is scientifically sound, and the results in principle may deserve the publication in MDPI Sensors journal. However, in the present form I cannot recommend this paper for the publication because of the English of the paper which should be substantially improved and polished with the help of a native speaker; sometimes it is hard to follow the authors’ logic. Moreover, there are several points that have to be properly addressed by the authors.
Response: First of all, thank you for your approval. We are very sorry for obscure expression in this paper, we have re-read the paper and have corrected the English expression with the help of a native speaker. Revisons in the full paper have been marked in red. Thank you for your comments.
Comment 2: Abstract is a simply replica of the Conclusion section; this is in some way bad manners; The research motivation presented in the Introduction is unclear and should be strengthen;
Response: Thank you for your guidance. We are sorry for the manner, in response to this question, we have rewritten Abstract and Conclusion; In Introduction, we have added the research motivation on L49-L53 in Section 1.
Comment 3: What is “a three-ring hexagonal lattice” (page 2)? May be the authors meant a circularly arranged hexagonal lattice?
Response: We are sorry we didn’t express clearly. We have corrected that the word “three-ring” on Section 2 is revised as “three-layer”. Thank you for your comments.
Comment 4: Here, the plasma frequency and electron frequency are given without units (Hz).
Response: We are sorry for our negligence here. The plasma frequency and electron frequency have been given with units (Hz) in Section.2. Thank you for your comments.
Comment 5: More details of the computer simulations will be welcome (i.e., COMSOL version, model dimension and type (full-wave or scattered field), light wave polarization;
Thank you for your suggestions. In the paper, the COMSOL 5.2 is adopted; the diameter of the PCF is 15μm; The source is an electromagnetic hybrid wave that is perpendicular to the x-y plane, and the full-wave analysis method is utilized to investigate the mode distribution. We have added relevant information in Section 2.
Comment 6: Page 3, the designations for the confinement loss alpha[loss] are different in eqs. (3) and (5).
Response: We are sorry for the mistake. We have unified the representation of the confinement loss in eqs. (3) and (5). Thank you for your comments.
Comment 7: I did not understand what are the “2-SPP” modes, please explain. How the eigenmodes of different polarizations (even, odd) were distinguished in the simulations?
Response: Thank you for your comments. We have added figure, station and corresponding literatures to explain 2-SPPmode. The contents are added in lines 110-116; The parity of the feature mode can be distinguished in the simulations according to the direction of the arrow, the content of this part is added in lines 120-123. We have added relevant references and marked red.
Comment 8: In fig. 3 the first and next to last data in the legend are marked identically. What is the meaning of the symbols (points) in this graph? If they indicate the specific values where the calculations were done so they should become denser in the region of function maximum.
Response: We are sorry for the mistake of identical mark. Fig.3 has been updated and in order to indicate the specific values, the points become denser in the region of peak maximum. Thank you for your comments.
Comment 9: In eqs. (6) and (7) the units (um) are missing in the right side;
Response: Thank you for your guidance. We have added the units (um) in the right side of eqs. (6) and (7).
Comment 10:Regarding fig. 8 I do not agree with authors that the tolerance of proposed sensor to the pores pitch variations (lambda) is as good as reported. Indeed, when examining figure 8b one can see that the change in lambda by -3% can produce the changes in the confinement losses of about 4% at 930 nm illumination wavelength! This should be discussed in more details.
Response: Thank you for your comments. Your comments are valuable and helpful for us. Our calculations show such results, we think and discuss carefully on your comments. And we add some explanations in the revised manuscripts for the possible causes. The possible reason for the result is that as the pitch decreases, the pores become denser, and the pore modulating effect on the filled core is more pronounced, so that the changes of energy and the drift of resonance wavelength are generated.
In addition, all expressions in the paper are in italics and we also added Shanghai Institute of Intelligent Science and Technology, Tongji University, 200092 Shanghai, China.
Round 2
Reviewer 1 Report
The authors have made a substantial effort to the last version, I would suggest a publication with some minor modifications mainly on the improvement of figures.
There is no colorbar in Figure 2, please consider to put it on. The arrows of the electric field do not show clearly. Please maybe use large arrows with a different color, e.g., black. In Fig. 3, please unify the font sizes for all, a, b and c subplots. In Fig. 5 (b), the inset a is slightly deformed. Please minorly change the aspect ratio.
Author Response
Comment: The authors have made a substantial effort to the last version, I would suggest a publication with some minor modifications mainly on the improvement of figures. There is no colorbar in Figure 2, please consider to put it on. The arrows of the electric field do not show clearly. Please maybe use large arrows with a different color, e.g., black; In Fig. 3, please unify the font sizes for all, a, b and c subplots; In Fig. 5 (b), the inset a is slightly deformed. Please minorly change the aspect ratio.
Response: Thank you for your suggestions. In Fig.2, we have added a colorbar, then large arrows with white have been marked in the electric field; In Fig.3, we have unified the font sizes for all, a, b and c subplots; In Fig.5(b), we have changed the aspect ratio to make the inset visible. Thank you for your approval and guidance again.
Reviewer 2 Report
The authors have addressed all my points and concerns. The work can be published in present form.
Author Response
Thank you for your approval and guidance again.